# Topological Map Construction Based on Region Dynamic Growing and Map Representation Method

**Fei Wang \*, Yuqiang Liu, Ling Xiao, Chengdong Wu and Hao Chu**

Faculty of Robot Science and Engineering, Northeastern University, Shenyang 110819, China; yuqiang0616@163.com (Y.L.); 1870719@stu.neu.edu.cn (L.X.); wuchengdong@mail.neu.edu.cn (C.W.); chuhao@mail.neu.edu.cn (H.C.)

\* Correspondence: wangfei@mail.neu.edu.cn; Tel.: +86-139-400-58702



**Featured Application: It is suitable for robots in the human-machine collaboration category, especially service robots.**

**Abstract:** In the human–machine interactive scene of the service robot, obstacle information and destination information are both required, and both kinds of information need to be saved and used at the same time. In order to solve this problem, this paper proposes a topological map construction pipeline based on regional dynamic growth and a map representation method based on the conical space model. Based on the metric map, the construction pipeline can initialize the region growth point on the trajectory of the mobile robot. Next, the topological region is divided by the region dynamic growth algorithm, the map structure is simplified by the minimum spanning tree, and the similar region is merged by the region merging algorithm. After that, the parameter TM (topological information in the map) and the parameter OM (occupied information in the map) are used to represent the topological information and the occupied information. Finally, a topological map represented by the colored picture is saved by converting to color information. It is highlighted that the topological map construction pipeline is not limited by the structure of the environment, and can be automatically adjusted according to the actual environment structure. What's more, the topological map representation method can save two kinds of map information at the same time, which simplifies the map representation structure. The experimental results show that the map construction method is flexible, and that resources such as calculation and storage are less consumed. The map representation method is convenient to use and improves the efficiency of the map in preservation.

**Keywords:** human–machine interactive navigation; mobile robot; topological map; regional growth

## 1. Introduction

Human–machine interactive navigation refers to the process through which machines and operator cooperate with each other to control the movement of devices and realize interactive navigation [1]. This type of navigation has been widely used in indoor service robots [2,3], self-driving cars [4,5], automatic guided vehicles (AGV) [6,7], and so on [8,9]. Among them, the most important part between the operator and machines is the map. The map needs to combine continuous spatial environment information with human abstract intentions, and discover the relationships between intentions and real spatial areas; finally, these connections are abstracted into a sequence of events (that is, a topological map) [10].

Many researchers began to study topological maps very early. Nodes are jointly represented by the sector feature of a laser and a proportional invariant feature of vision, which do not depend on

any artificial landmark, and the global location of robots in the process of map creation [11]. However, the laser sector feature is required in the intersection area among different channels, and the visual feature is also required to be proportional invariant. Therefore, the application effect will be affected in such large-scale and featureless scenarios as a living room. The final topological map is sparse, and cannot achieve accurate navigation obstacle avoidance [12]. A self-organizing method of hierarchical clustering (Map-TreeMaps) is proposed in [13]. Each unit of the map represents the structured data of the tree, while the treemap method provides a global view of the local hierarchy. This method enhances the generality of the construction of topological maps and solves the problem that some environmental geometric structures are constrained. Similar to the Chow–Liu tree model in [14], as long as the number of searching layers is big enough, various spatial structures can be detected and clustered separately. In addition, the segmentation results can be easily optimized by thresholding the weights of local subgraphs. Both methods abandon the details of the local subgraph, which makes it difficult to achieve accurate navigation obstacle avoidance. An auxiliary graph is used to solve the problem of local subgraph association in [15], which improves the efficiency of segmentation and doesn't solve the problem of subgraph details.

Moreover, the above clustering segmentation cannot deal with large-scale and open space, which is not human-friendly for human–machine interactive navigation. For example, in the application of service robots, large living rooms and long corridors cannot be regarded as a region, but should be divided into several regions according to the actual situation [16]. A novel and efficient method for updating Voronoi diagrams was proposed in [17], which only updates those units that are actually affected by the environment, and finally lower the number of visits and computing time. In addition, a skeleton-based Voronoi diagram method is also proposed, which is particularly effective for noise removal. A simultaneous location and mapping algorithm (VorSLAM) based on Voronoi map representation is proposed in [18]. One of the basic features of this algorithm is that the features correspond to the local map one by one, and each feature is associated with a local map defined on the feature. This not only retains the details of the local map, it also alleviates the problem of large scene segmentation by Voronoi partitioning. However, the Voronoi graph is based on the principle of distance or special structure to divide the space, so that the area obtained is "basically the same size", and the special structure also limits the applicability of the algorithm.

From the research history of topological map construction, a generalized Voronoi map [19] and spectral clustering [20] are the two main methods for topological map construction at present [21]. For example, a lightweight method is proposed to create maps by combining metric maps and topological information in [22]. By combining the information of two maps, the robot can realize autonomous navigation and obstacle avoidance in a large area. In paper [23], spectral clustering and an extended Voronoi graph are used to construct a topological graph from a metric graph. The specific idea is to use spectral clustering to segment the metric graph and get the center of the cluster. After determining the first vertex, other vertices are established by an extended Voronoi graph, and vertices are divided into connection points and tail nodes. Although the combination of two maps achieves navigation avoidance, the difficulty of map preservation is increased; the combination of two segmentation methods improves the efficiency and scope of application of topological segmentation, and the redundancy and uncontrollability of segmentation results is coming, which easily leads to the accumulation of topological vertices in some areas.

To solve the above problems, this paper proposes a novel pipeline of constructing a topological map and an efficient way of expressing a topological map. The main contributions of this paper are as follows:

- A complete system of building, saving, and loading topographic maps is proposed, which can make the topographic maps readily applied.
- A topological segmentation method based on region dynamic growth is proposed, which makes the region segmentation no longer limited by the geometrical structure of the environment, and also more in line with the actual needs of human–machine interactive navigation.

- A representation method of a topological map based on the conical space model is proposed, which makes the map retain not only the information of the topological relationship, but also the information regarding the obstacle occupied.

Finally, several comparative experiments are carried out in the Gazebo simulator provided by ROS (Robot Operating System, ROS) and the author's lab. The experimental results verify the effectiveness of the proposed system and method. The overall block diagram of the topological map building system is shown in Figure 1.

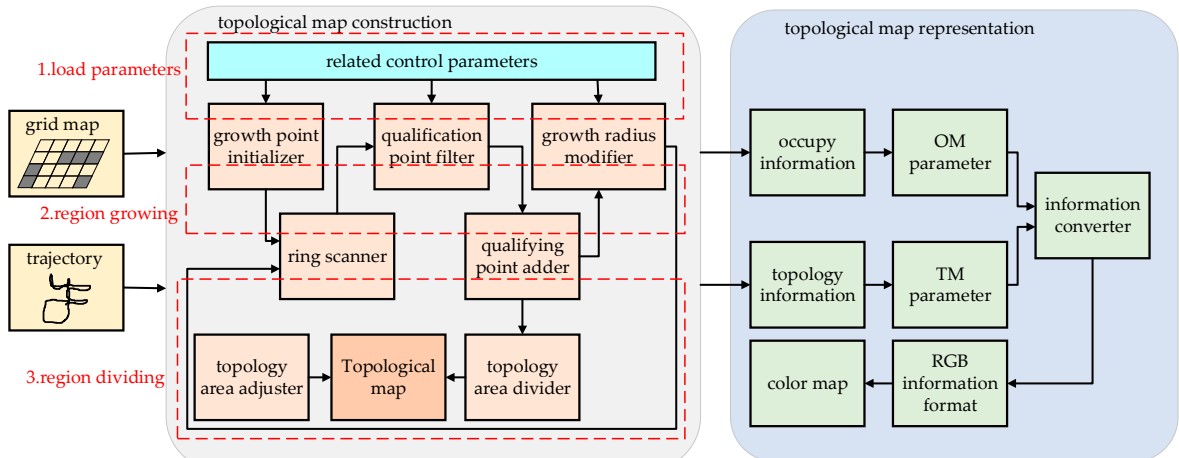

**Figure 1.** The architecture of the topological map construction system.

## 2. Topological Map Construction Based on Regional Dynamic Growth

The process of building a topological map based on region dynamic growth is shown in the left part of Figure 1. The system first randomly selects the initial growth point on the trajectory of the robot, and then grows dynamically according to the control parameters. When the regional characteristics meet the requirements, the growth will stop. This method is inspired by the spherical expansion of voxel clusters in paper [24]. Then, after each region has grown, it will be segmented from the meter map, and the color identification of the topological region will be established. The center of gravity of the topological region will be used as the center of the region, and the neighborhood of the region will be scanned. Finally, after all the topological areas have been established, the region adjustment part will delete, merge, and grow the unqualified areas, and finally complete the construction of the whole environment's topological map.

### 2.1. Regional Growth Process

Based on the metric map and the trajectory of the robot, the map building system will randomly initialize the region growing point on the trajectory (which is in the same coordinate system as the metric map). It is worth mentioning that this point is not the final center of the region (also called the topological vertex). Then, the circular growth of the region is achieved by using a circular scanner with the same degree of growth in all directions on the plane, rather than depending on the geometric structure or special characteristics of the environment.

There are several important parameters in the regional dynamic growth algorithm, which are described below:

- $r_a$: The points' addition ratio in the circular scan, which refers to the ratio of qualified points in each circular scan. It can prevent malformation, making the growth area approximately circular or elliptical rather than elongated.
- $r_o$: The obstacle ratio in the circular scan, which refers to the ratio of unqualified points to the total qualified points in the convex hull region composed of qualified points in the added points

and areas in each annular scanning. It can prevent the region from growing into a concave region, and ensure the convexity of the topological region.

■ $r_p$: The points pass ratio of the topological region, which refers to the ratio of all the points in the topological region to all the points in the circular region (ideal region). It can reflect the contrast between the growth area and the ideal area in the current state. It is used to modify the growth radius in real time, so its role is to prevent growth deficiency.

■ $R_w$: The control weight of dynamic modification of the growth radius, which refers to the extent of radius modification in each region.

The concept of the qualified point is mentioned above. If one of them is not satisfied, it is called an unqualified point. The qualified point needs to satisfy all the following conditions:

1. The metric map area that corresponded to the point must be in a free space, and cannot be an occupied space or an unknown space.
2. The topological map area corresponding to the point must be a non-topological identifier area, which can only be a free space or a local area identifier.
3. Conditions 1 and 2 must be satisfied for all points through which the ray emitted from the region vertex passes.

As the location of the robot's trajectory must be free and can basically traverse the whole environment, it is better to sample the growing points from the robot's trajectory. Then, the main steps of the region dynamic growth algorithm are as follows (Figure 2, the symbols in the Figure 2d will be described later):

(1) Determine the growth point and scan all the adjacent points around the current region vertex in a circular way.
(2) Rays from the growth point to each point were calculated using the Bresenham algorithm [25], and all of the qualified points were screened.
(3) Calculate the convex hull after each qualified point is added using the Graham scanning algorithm [26], and retain the point where the obstacle ratio satisfies the requirements.
(4) Add qualifying points to the topological area. At the same time, the points pass ratio and the points add ratio in the current state are calculated. If the points' addition ratio is less than the threshold, the growth of the current region is exited, indicating that the region growth is completed ahead of time. If the addition rate is normal, the points pass ratio will be used to modify the maximum growth radius. The modified formula is shown in Equation (1):

$$R_{\max}^{t+1} = R_{\max}^t + R_w \cdot r_p \cdot R_{\max}^t \tag{1}$$

where $R_{\max}^t$ is the maximum growth radius at the current time $t$, and $R_{\max}^{t+1}$ is the maximum growth radius at the time $t + 1$.

After that, the regional dynamic growth algorithm will grow in other areas in the same way until the topological area covers most of the environment.

The center of each region will be updated after the end of each region growth. This paper considers that every topological region is an irregular convex polygon composed of convex hull points in the region, and the polygon is approximately circular or elliptic, so the center of gravity of the polygon can be used as a new center.

As shown in (d) of Figure 2, the polygon is considered to be composed of multiple triangles of area $S(S_0, S_1, S_2, \ldots)$, where $p_c$ represents a vertex and $p_0 p_1, p_1 p_2, p_2 p_3, \ldots$ represent the other two points in triangles. Then, if the coordinates of three vertices of the triangle $\triangle p_1 p_2 p_3$ are known: $p_1(x_1, y_1), p_2(x_2, y_2), p_3(x_3, y_3)$, the common center of gravity coordinates can be obtained, as shown in Equation (2).

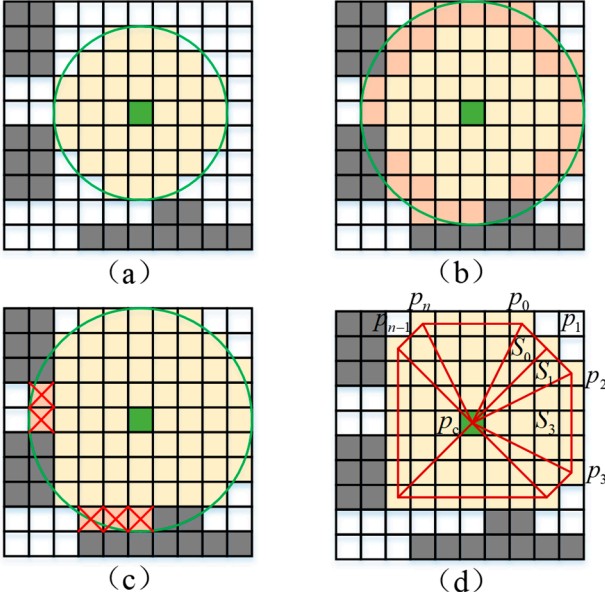

**Figure 2.** Main steps of the regional dynamic growth algorithm. (**a**) Growth state at a certain moment. (**b**) The next moment is grown in a circular expansion manner to obtain adjacent candidate points. (**c**) Remove the unqualified points in the adjacent candidate points. (**d**) Add qualified candidate points to the topological region.

However, Equation (2) is not suitable for multi-triangle calculation, so the triangle area calculation method is used in this paper. Firstly, the area of the triangle is calculated by vector cross-multiplication, as shown in Equation (3). In addition, it is not necessary to consider whether the traversal order of the three points is clockwise or counter-clockwise in the calculation process, because it can be cancelled in the subsequent calculations.

$$
\begin{cases}
x_g = \frac{x_1 + x_2 + x_3}{3} \\
y_g = \frac{y_1 + y_2 + y_3}{3}
\end{cases}
\tag{2}
$$

$$
S = \frac{(x_2 - x_1) * (y_3 - y_1) - (x_3 - x_1) * (y_2 - y_1)}{2}
\tag{3}
$$

Assuming that an irregular convex polygon is composed of $n$ triangles, in which each triangle has an area of $S_i$ and a center of gravity of $G_i(x_i, y_i)$, the integral can be transformed into an accumulation sum by using the formula for calculating the center of gravity of a planar thin plate, as shown in Equation (4):

$$
\begin{cases}
x_g = \dfrac{\iint_D x \, dS}{S} = \dfrac{\sum\limits_{i=1}^{n} x_i S_i}{\sum\limits_{i=1}^{n} S_i} \\[4mm]
y_g = \dfrac{\iint_D y \, dS}{S} = \dfrac{\sum\limits_{i=1}^{n} y_i S_i}{\sum\limits_{i=1}^{n} S_i}
\end{cases}
\tag{4}
$$

Then, in any irregular convex $n$ polygon $(p_0, p_1, p_2, \ldots, p_n)$ with $p_i(x_i, y_i)(i = 0, 1, 2, 3, \ldots, n)$ as a vertex, it is divided into $n$ triangles composed of a center point $p_c(x_c, y_c)$ as a vertex and any two points $p_i$. Then, the coordinates of the center of gravity of the polygon are calculated as shown in the following Equation (5):

$$
\begin{cases}
x' = \dfrac{\sum\limits_{i=1}^{n}(x_c+x_i+x_{i-1})S_i}{3\sum\limits_{i=1}^{n}S_i} \\[4ex]
y' = \dfrac{\sum\limits_{i=1}^{n}(y_c+y_i+y_{i-1})S_i}{3\sum\limits_{i=1}^{n}S_i}
\end{cases}
\tag{5}
$$

Finally, the principal semi-axial length and the short semi-axis length of the region are calculated by principal component analysis. The pseudo code of the regional dynamic growth algorithm is shown in Algorithm 1.

---

**Algorithm 1:** Regional dynamic growth

---

**Objectives:** The growing point is regarded as a temporary topological vertex, and then all the neighboring points are scanned annularly to select eligible points and add them to the current topological region.
**Input:** Growth point $p_c(x, y)$, maximum growth radius $R_{max}$.
**Output:** Central point $p'_c$ of the topological region, radius $R'_{new}$ of the topological region.
1:    initial growth radius $r$
2:    initialization of eligible point set $P_p = \{\}$
3:    initialization of unqualified point set $P_f = \{\}$
4:    **for** $(r = 1; r < R_{max}; r++)$
5:        $P_t = \{\}$
6:        $P_t \leftarrow$ ring scanner $(p_c, r)$
7:        **for each** $p_i$ **in** $P_t$
8:            **if** ($p_i$ is failure)
9:                $P_f \leftarrow$ fail point filter $(p_i)$
10:               **continue**
11:           $P_r \leftarrow \{\}$
12:           $P_r \leftarrow$ calculate the point at which ray $(p_c \rightarrow p_i)$ passes
13:           **if** ($P_r$ is failure)
14:               $P_f \leftarrow$ fail point filter$(p_i)$
15:               **continue**
16:           $P_h \leftarrow$ calculate the current convex hull $(P_p, p_i)$
17:           calculate the number of unqualified points containing $P_f$ in convex hull $P_h$
18:           **if** (obstacle ratio $r_a$ doesn't satisfies the requirements)
19:               $P_f \leftarrow$ fail point filter $(p_i)$
20:       $P_p \leftarrow P_t$, and add a topological area identifier
21:       **if** (addition rate $r_a$ satisfies the requirements)
22:           $R'_{max} \leftarrow$ modify the growth radius $(R_{max}, r_p, R_w)$
23:           $R_{max} \leftarrow R'_{max}$
24:       **else break**
25:   $P'_c \leftarrow$ recalculate the regional center $(P_c, P_p)$
26:   $R'_{new} \leftarrow$ recalculate the radius information of the area $(P_p)$
27:   **return** $R'_{max}, P'$

---

*2.2. Regional Adjustment Process*

After the growth of the previous section, the whole environment will be divided into regions of different sizes because of the dynamic growth. The small areas not only waste the vertex resources of the topological map, but also cannot be used for the interactive navigation of the actual scene. Therefore, it is necessary to merge small areas and optimize the topological map.

In order to improve the coverage of topographic maps, the region can be regenerated. During the secondary growth process, if the radius of the current region is less than the threshold value and no

new points are added, or the maximum radius of secondary growth is meeted, the growth process will quit. The method is similar to regional growth in Section 2.1, which is not discussed here again.

In addition, the Kruskal algorithm is used to adjust the topological map before region merging, and the minimum spanning tree form of the topological map is obtained, which further simplifies the map structure. The next step is to merge some areas. The region merging algorithm first counts the topological vertices that meet the merge requirements. It needs to satisfy the following two conditions:

1.  The radius of the main vertex area is less than the threshold.
2.  The total area of the merged area is less than the threshold.
3.  The obstacle ratio of the merged area is less than the threshold.

---

**Algorithm 2:** Region Merging

---

**Objectives:** To merge small areas, delete the original topological vertices and generate new topological vertices, and change the color identification of the topological area.
**Input:** Topological region vertex sequence set $P_m$ to be merged.
**Output:** The status of this subarea merge: true means the merge was successful; false means no merge occurred.

1:  initialize the set of topological vertex ordinals $V_{com} = \{\}$ that need to be merged
2:  initialize the point set $P_{com} = \{\}$ of the points contained in the merged region
3:  initialize the point set $P_f = \{\}$ of the disqualified points around the merged area
4:  initialize the first vertex information ($p_{first} \leftarrow P_m^0$) of the merge process
5:  calculate the weighted center coordinate $p_c \leftarrow P_m$
6:  calculate the average scan radius $r_c \leftarrow P_m$
7:  **for** $r$ **in** $[0, r_c]$
8:      $P \leftarrow$ loop traversal of all the points contained within the scan radius.
9:      **for** $P_i$ **in** $P$
10:         **if** ($P_i$ belongs to the topological area)
11:             $P_{com} \leftarrow P_i$
12:         **else** $P_f \leftarrow P_i$
13: **if** ($P_{com}$ is empty)
14:     **return false**
15: $P_h \leftarrow$ calculate the current convex hull ($P_{com}$)
16: calculate the number of unqualified points containing $P_f$ in convex hull $P_h$
17: **if** (obstacle ratio $r_a$ doesn't satisfies the requirements)
18:     **return false**
19: delete all of the vertex information in $P_m$
20: Merge the vertex regions in $P_m$ and add the region identifier at the same time
21: $P_c' \leftarrow$ recalculate the center of the area
22: $r_n \leftarrow$ recalculate the radius of the merged area
23: Add a new topological vertex ($r_n, P_c', P_{com}$)
24: **return true**

---

After the vertex numbers that need to be merged are obtained, the region merge can be performed. The main steps are as follows:

(1)  The points in the corresponding topological region of all the vertices are extracted, and the qualified points (identical with the vertex color marker) and the failure points (different from the vertex color) are counted.
(2)  The convex hulls of qualified points are calculated, the unqualified points of convex hulls are counted from the failure points, and the obstacle ratio is calculated. If the obstacle ratio is greater than the threshold, then exit.
(3)  If the obstacle ratio is less than the threshold, the region merging is carried out, the original vertex is deleted, and a new vertex is established.

The pseudo code of the algorithm is shown in Algorithm 2.

## 3. Topological Map Representation

The process of map preservation is shown in the block diagram on the right side of Figure 1, from which it can be seen that the representation method of the topological map is a process of combining the metric map with the topological map. The method is inspired by the ROS package [27], which expands from a gray value to an RGB color, so that the occupied information and the topological information is saved efficiently while saving and reusing the topological map. In this paper, the parameter TM and the parameter OM are used to represent the topological information and the occupied information, respectively; then, the information converter is used to convert the parameters into the RGB value of the color picture, so that the obtained RGB value can not only be used to display and identify the topology area in real time, but can also contribute to save the map information as a color image.

### 3.1. Online Representation of the Map

The metric map divides the space into a finite number of grid cells $M= \{m_i | i = 0, 1, 2, \ldots n\}$, each $m_i$ corresponding to an occupied variable. If the grid cell is completely occupied by "1" and is not occupied as "0", then $p(m_i = 1)$ or $p(m_i)$ indicates the possibility that the grid cell is occupied [28]. Therefore, the smallest unit of the metric is the grid cell, and a grid cell is represented by only one variable.

This only represents the information-occupied obstacles; although it can achieve accurate navigation and obstacle avoidance, it cannot achieve a higher level of control. For example, the user tells the robot "I want to go to the kitchen!", and the robot does not know how to perform unless the robot knows the specific location of the kitchen [29]. In order to solve this problem, a scheme combining the metric map and the topological map has been proposed.

A topological map is an environment representation method based on an adjacent graph, which can be represented by an adjacency list structure. Similar to the concept of graphs, topological maps have two basic elements: nodes (which are called vertices in this paper) and edges. Nodes represent different locations that can be distinguished in the environment or various states of distinguishable robots; edges represent relationships between nodes, such as distance or motion control commands. Such representations have low storage requirements, and also support efficient path planning, especially for large-scale unknown environments or outdoor environments.

In this paper, the vertices mainly include the position of the vertices, the vertex number, the adjacent vertices, and so on. The edges mainly include starting and ending vertices, weights, etc. In addition, metric maps and topological maps exist in the form of a "hierarchical map" in the framework proposed in this paper. The two maps interact with each other, learn from each other, and complement each other, providing different map services for the system.

### 3.2. Topological Map Preservation

The preservation of topological map is the process of integrating topological information and occupied information to form RGB image information. In order to describe the two kinds of map information conveniently, a conical space model is proposed, which is controlled by the parameter TM, the parameter OM, and integer one. The height of the cone is determined by the integer one, and the height is always one. The radius of the bottom surface of the cone is determined by the parameter OM, and the value of the polar coordinates of the bottom surface is determined by the parameter TM, which is inspired by the HSV color space theory [30]. An example process is shown in Figure 3 below.

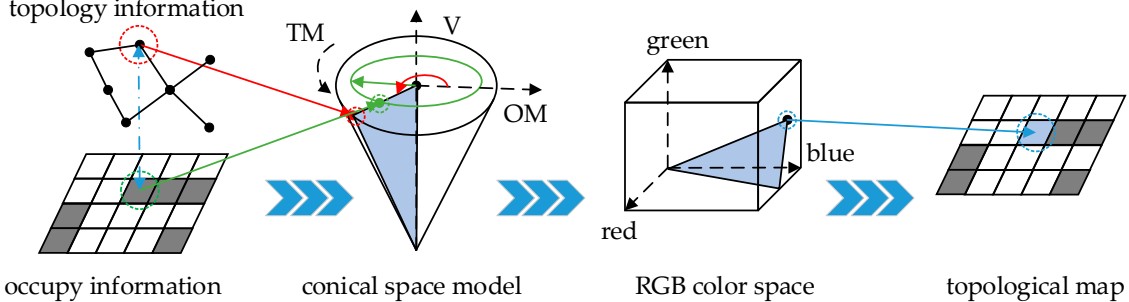

**Figure 3.** Topological map preservation process.

In the figure, the red line represents the conversion process of the topological information, the green line represents the conversion process of the occupied information, and the blue line represents the synthesis process between the two pieces of information. It should be noted that the relationship between the grid cell and topological vertex is "many-to-one", which means that a topological vertex corresponds to multiple grid elements. In other words, multiple grid cells form a region, and a region is represented by a topological vertex.

In the process of preservation, the information of the topological vertex is extracted from the topological map firstly; then, the occupied information of the grid cell corresponding to the topological vertex is found in the metric map, and the two pieces of information are converted into the parameters TM and OM. In the conical space model, a cross-section can be obtained by these two parameters. The cross-section represents the result of the synthesis of two kinds of information. However, such information is still unusable, and needs to be converted again. The blue triangle cross-section is shown in Figure 3. Therefore, the information converter is proposed to convert the cross-section information into RGB information, which is often used in the field of vision. The RGB value is used to represent a grid cell, and then each grid cell is operated in the same way. Finally, the color map containing the topological information and occupancy information is saved. In the following, the TM and OM parameters are explained in detail:

TM is measured by the angle, and the range of values is [0, 360]. It is calculated counterclockwise from red. This value is used to represent the topological neighborhood, which can represent at most 360 topological neighborhoods. In order to show a higher degree of discrimination, values can be taken at intervals, such as 20 color values for the identification of the topological region, and 18 kinds of topological region identifiers can be used.

OM represents the degree of proximity to the spectral color, and the range of values is [0, 1]. This value is used to represent whether the obstacle is occupied at a point in the map. The larger the value, the greater the likelihood that there is an obstacle at that point; the smaller the value, the greater the likelihood that the point may be free.

Then, in the beginning of topological map preservation, the occupied information and topological information need to be converted. The conversion formulas are as shown in Equation (6):

$$\begin{cases} TM = a + b \times P_i \\ OM = k \times p(m_i) \end{cases} \tag{6}$$

where $a$ is the start value, $b$ is the interval value, $k$ is the certain coefficient, and $a = 0, b = 20, k = 1$ is taken in the paper.

After that, some intermediate variables should be computed from these two parameters, and the formulas are shown in Equation (7), where $V = 1$ is taken in the paper. Then, with the help of intermediate variables, the RGB information can be obtained by Equation (8):

$$
\begin{cases}
h = \left\lfloor \frac{TM}{60} \right\rfloor \quad (\mathrm{mod}6) \\
f = \frac{TM}{60} - h \\
p = V \times (1 - OM) \\
q = V \times (1 - f \times OM) \\
g = V \times (1 - (1 - f) \times OM)
\end{cases}
\tag{7}
$$

In addition, the yaml file and the picture in pgm format are used to save the metric map traditionally, where the picture in pgm picture is a grayscale picture, and only the gray level data can be saved. Therefore, the picture in ppm picture is used to save the topological map, because the picture in ppm format can save RGB color data. Moreover, the ppm image format is divided into the ASCII encoding format (file descriptor is P3) and the binary-encoding format (file descriptor is P6), because the binary encoding format consumes less memory than the ASCII encoding format, so the binary encoding format is used.

$$
(R, G, B) =
\begin{cases}
(255, g, p) & \text{if } h = 0 \\
(q, 255, p) & \text{if } h = 1 \\
(p, 255, g) & \text{if } h = 2 \\
(p, q, 255) & \text{if } h = 3 \\
(g, p, 255) & \text{if } h = 4 \\
(255, p, q) & \text{if } h = 5
\end{cases}
\tag{8}
$$

Except for the ppm image, a yaml file that contains the pixel coordinates of topological vertices, adjacent vertices, and other information is also saved, and the details saved in this yaml file will be explained in the next section. In order to simplify the calculation, in the following experiments, the area indicated by red is occupied, the area indicated by green is free and the area indicated by blue is unknown.

### 3.3. Topological Map Reading

The reading of the topological map means that the map file is parsed first, and then the obstacle occupied information and topological information are restored. The reading process takes two steps: reading the yaml file and the ppm image.

Here, to explain the contents of the yaml file, the file contains the following main parts:

◆　Image file path: Refers to the saved path of the ppm image file.
◆　Resolution: Refers to the resolution of the map, and is used to represent the scale of a pixel in the real world, with a unit (meters/pixel).
◆　Origin: Refers to the two-dimensional (2D) pose of the lower left pixel in the map, as (x, y, yaw), with yaw as the counter-clockwise rotation (yaw = 0 means no rotation). The yaw is ignored in this paper.
◆　Free thresh: Pixels with an occupancy probability less than this threshold are considered completely free.
◆　Occupied thresh: Pixels with an occupancy probability greater than this threshold are considered completely occupied.
◆　Mode: The way the file is saved, which can be one of three values: trinary, scale, or raw. Trinary is the default.
◆　Number of topological vertices.

◆　　Topological information: A series of lists that contain the pixel coordinates of topological vertices, adjacent vertices, and other information (such as semantic information; this is empty in the paper).

In the map representation method proposed in this paper, the metric map is a two-dimensional map, so the map coordinates, pixel coordinates, initial points, and other relationships can be shown in Figure 4 below. In the image, different color regions represent different topological regions, and dark color regions are more likely to be occupied.

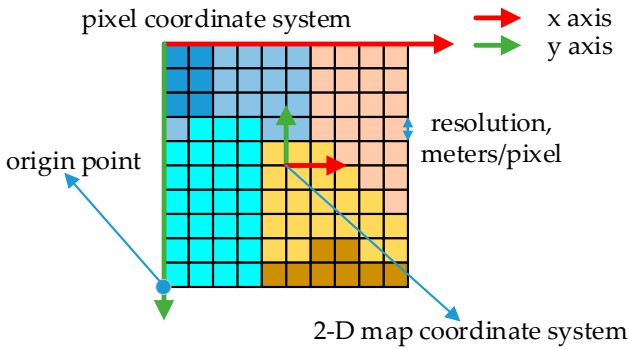

**Figure 4.** Relations between coordinate systems in the picture in ppm format.

Traditionally, the origin of the pixel coordinate system is in the upper left corner of the picture, while the origin of the map coordinate system is in the center of the picture, and the pixel coordinates of the center can be calculated by the origin point. Then, with the help of the resolution of the map, all the pixels can be restored to grid cells.

In the first place, the pixel coordinates of the pixel points in the picture need to be converted into map coordinates by the map coordinates of the lower left pixel, and the specific conversion formula is as shown in Equation (9):

$$\begin{cases} x_{\mathrm{map}} = x_{\mathrm{orign}} + x_{\mathrm{pixel}} * res \\ y_{\mathrm{map}} = y_{\mathrm{orign}} + h_{\mathrm{map}} * res - y_{\mathrm{pixel}} * res \end{cases} \tag{9}$$

where $(x_{\mathrm{pixel}}, y_{\mathrm{pixel}})$ is the pixel coordinate, $(x_{\mathrm{map}}, y_{\mathrm{map}})$ is the map coordinate, *res* is the map resolution, and $h_{\mathrm{map}}$ is the height of the map.

At the same time, the RGB image information in the ppm image file needs to be converted into two parameters in the conical space model, and the conversion formula is shown in Equation (10).

$$\begin{aligned} V &\leftarrow \max(R, G, B) \\ OM &\leftarrow \begin{cases} \frac{V - \min(R,G,B)}{V} & \text{if } V \neq 0 \\ 0 & \text{otherwise} \end{cases} \\ TM &\leftarrow \begin{cases} \frac{60(G-B)}{V-\min(R,G,B)} & \text{if } V = R \\ \frac{120 + 60(B-R)}{V-\min(R,G,B)} & \text{if } V = G \\ \frac{240 + 60(R-G)}{V-\min(R,G,B)} & \text{if } V = B \end{cases} \end{aligned} \tag{10}$$

where if $TM < 0$, then $TM = TM + 360$. After the above calculation, the final range of the three values is: $0 \leq V \leq 1, 0 \leq OM \leq 1, 0 \leq TM \leq 360$.

Afterwards, the topological information can be obtained from the parameter TM, and the topological map is established. In addition, the occupied information of the point is read from the parameter OM, and the metric map is established. In this way, the topological information and occupied information can be completely recovered.

## 4. Experiment and Result

Finally, several experiments will be carried out to illustrate the effectiveness of the proposed algorithm and the rationality of the topological map representation method. Relevant experiments were completed in the simulation environment, and the metric map was obtained by using the gmapping algorithm [31].

In the Gazebo simulation environment, four environments were built representing an indoor home, office, pillar, and open space. The experimental environment is shown in Figure 5. In the experiment, the turtlebot2 robot equipped with a 2D laser sensor was used as an experimental platform. All of the experiments were performed on a notebook with a memory 8G, i7 processor, and a GTX1050 graphics card.

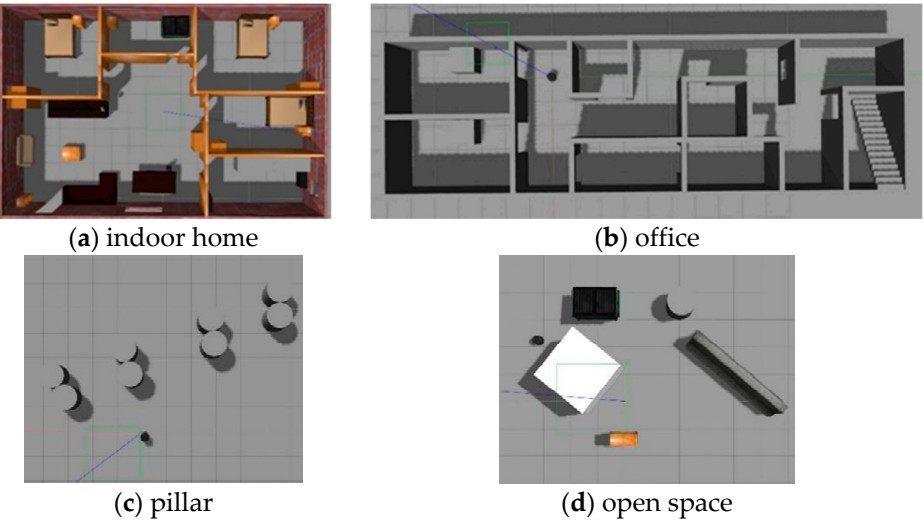

(**a**) indoor home      (**b**) office

(**c**) pillar      (**d**) open space

**Figure 5.** Simulation environment.

The first experiment was carried out in the indoor home environment, and each part of the whole system was tested. The experimental results are shown in Figure 6, which includes the construction of the metric map, the construction of the topological map, the adjustment of the topological map, and the preservation of the topological map. These continuous processes truly implement the process of topological mapping from build to save.

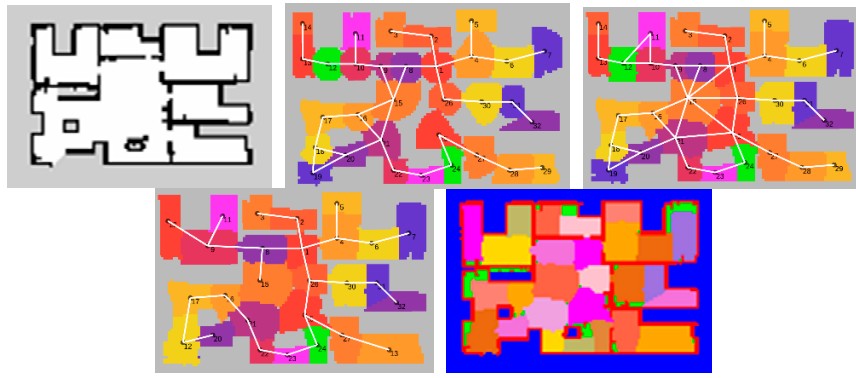

**Figure 6.** Topological map construction process.

In the following, relevant experiments were carried out in the office environment, pillar environment, and open space. The results are shown in Figure 7, which shows the metric map and topological map constructed in different environments. Among them, the growth radius is between 1.2–3 m, the map resolution is 0.20 m, and the control weight is one.

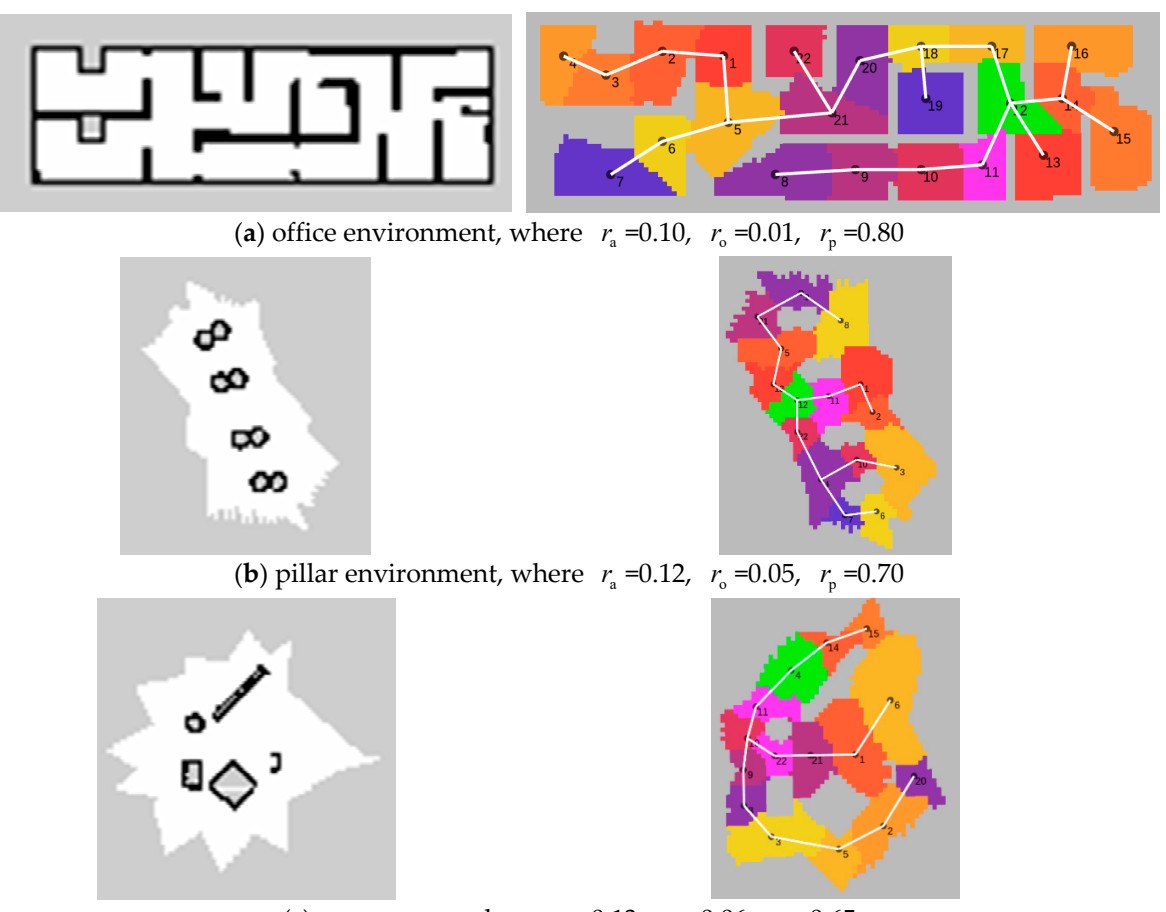

(**a**) office environment, where $r_a$ =0.10, $r_o$ =0.01, $r_p$ =0.80

(**b**) pillar environment, where $r_a$ =0.12, $r_o$ =0.05, $r_p$ =0.70

(**c**) open space, where $r_a$ =0.13, $r_o$ =0.06, $r_p$ =0.65

**Figure 7.** Occupied map (**left**) and its corresponding topological map (**right**).

The office environment consists of many small spaces and long corridors. It can be seen that the map construction algorithm performs poorly at the door and the area is somewhat deformed. However, in the latter two examples, the map construction algorithm is more perfect: the region is more similar to the ellipse, and the dynamic growth and merging process makes the small region fully covered.

Next, in the indoor environment, this paper chooses a special location (next to the dining table in the living room) to build the map, and discusses the impact of these three parameters on the map construction under the circumstances of changing the obstacle ratio, the point addition ratio, and the points pass ratio. Here, the growth radius is set to 1.5 m, the map resolution is 0.2 m, and the control value $R_w$ is set to one. The details of regional growth are shown in Figure 8.

The obstacle ratio is to prevent the region from growing into a concave region, and ensure the convexity of the topological region. As can be seen in group (**a**) of Figure 7, with the increase of the obstacle ratio, the area will grow toward a small space (such as a door) next to it, and a few obstacle points will be added continuously.

The points addition ratio prevents malformation, making the growth area approximately circular or elliptical rather than elongated. It can be seen in group (**a**) of Figure 8 that when the point addition ratio is too small or too large, it is easy to cause growth malformation or insufficient growth.

The points pass ratio can reflect the contrast between the growth area and the ideal area in the current state. It is used to modify the growth radius in real time, so its role is to prevent growth deficiency. When the growth area is far from the ideal area, it can make the region grow seriously. This effect is well reflected in the (**c**) group, but it also destroys the convex nature of the area. Therefore, the

above three parameters need to be adjusted according to the actual situation, and the topological map will represent the environment better.

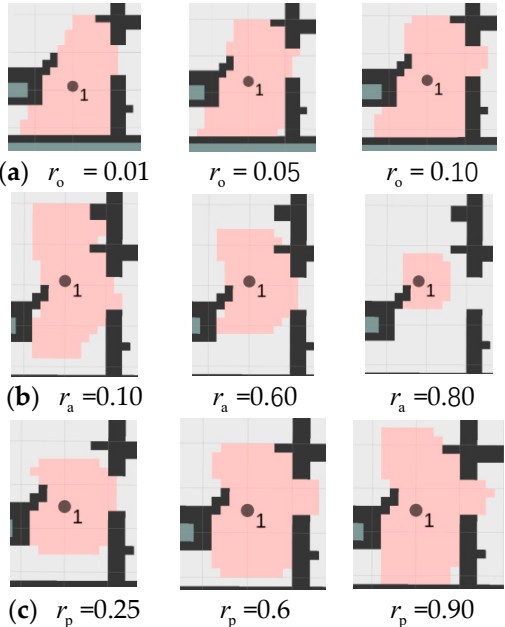

**Figure 8.** The influence of three parameters on the construction of the topological map. (**a**) Impact of changes in the obstacle ratio on the region. (**b**) Impact of changes in the point addition ratio on the region. (**c**) Impact of changes in the points pass ratio on the region.

In addition, map resolution is another important factor affecting the construction of topological maps. In the indoor home environment, a large number of experiments were carried out for different resolutions in order to obtain an optimal topological map. Figure 9 shows the effect of map resolution on topological map connectivity and occupancy.

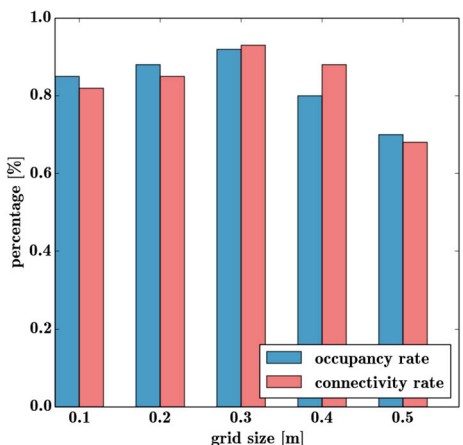

**Figure 9.** The impact of map resolution on topological map construction.

Finally, topological maps built in the office environment, pillar environment, and open space are saved, as shown in Figure 10. Experiments show that the topological map representation proposed in this paper can save the occupied information and topological information well.

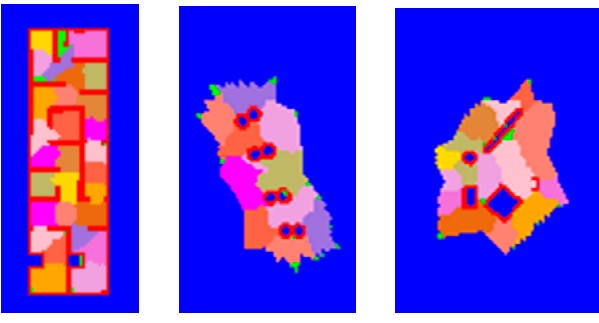

**Figure 10.** Topological maps saved in three environments.

What's more, the system proposed in this paper was integrated into the turtlebot2 equipped with the hokuyo laser sensor and NVidia TK1, and the topological map construction was successfully completed in the author's laboratory, refer to the Supplementary Materials. This experiment proves the practicality of the pipeline proposed in this paper on mobile platforms such as service robots. The configuration and map of the turtlebot2 experimental system is shown in Figure 11.

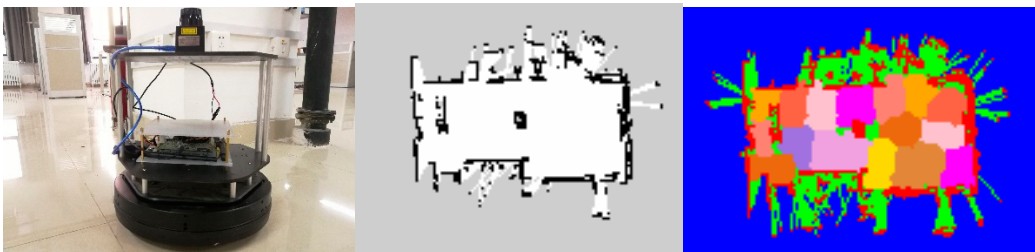

**Figure 11.** Setup of the turtlebot2 experiments and maps.

Not only is the topological map construction method not limited by the environment geometry, but the topological map preservation method can also simultaneously save the occupied information and topological information to achieve the map conditions required for accurate navigation. In order to better illustrate that, Table 1 summarizes some of the information about the topological map creation in various environments. As can be seen from the table, the topological map can cover the entire environment, and the more complex the environment, the more topological vertices the map construction method will use to describe it. Although the topological map storage requirements are larger than the metric maps, more information will be saved to provide the same precision navigation environment.

**Table 1.** Topological map information for each scene.

| | Topological Map Information | | Storage Requirements (kB) | |
|---|---|---|---|---|
| | **Number of Vertexes** | **Occupied Ratio** | **Metric Map** | **Topological Map** |
| Indoor (171.224 m$^2$) | 31 | 0.922444 | 36.0183 | 109.46 |
| Office (115.389 m$^2$) | 22 | 0.985809 | 30.0172 | 91.27 |
| Pillar (129.821 m$^2$) | 15 | 0.984625 | 20.0172 | 72.061 |
| Open space (171.261 m$^2$) | 15 | 0.936123 | 16.0172 | 48.0853 |
| Laboratory (277.999 m$^2$) | 16 | 0.883049 | 30.0181 | 90.0903 |

## 5. Conclusions

In this paper, a new framework for creating high-precision topological maps and efficient map representations for diverse environments is presented. In various structured environments, the regional dynamic growth algorithm proposed in this paper can divide the free space into multiple convex regions, each forming a topological region to represent the environment. This greatly reduces the use

requirements compared to the most advanced methods, and extends the range of use. The topological map representation method proposed in this paper uses the parameter TM and the parameter OM of the conical space model to represent the occupation information and topological information of the map efficiently, which satisfies the conditions of high-precision navigation.

Through a large number of experiments, the topological map construction method proposed in this paper can construct a topological map that conforms to a specific environment without being limited by the environment geometry. In addition, the topological map can be made to better describe the entire environment by modifying the corresponding control parameters. It also verifies that the topological map preservation method proposed in this paper can save the occupied information and topological information at the same time by occupying a small amount of storage space. The universality of the construction method and the efficiency of the preservation method provide conditions for human–machine interactive navigation, so that the topological map can be truly applied in real life.

For the future research, since this paper only studies two-dimensional topological maps, three-dimensional topological maps will be the direction of future work. In addition, the semantic information of the actual scene can be added to the topological node to improve the robot's understanding of the actual environment.

## 6. Patents

A patent named "Human–machine interactive navigation system and method based on brain–computer interface" is pending.

**Supplementary Materials:** A video is available online at https://youtu.be/XUidY4vnslU, Video title: A topology map construction experiment based on dynamic growth algorithm.

**Author Contributions:** This study was completed by the co-authors. F.W., C.W., and H.C. conceived and led the research. The major experiments and analyses were undertaken by Y.L. and L.X.; F.W. supervised and guided this study. Y.L. and L.X. wrote the paper. All the authors have read and approved the final manuscript. Conceptualization, F.W.; Formal analysis, Y.L.; Investigation, Y.L.; Methodology, Y.L.; Project administration, F.W., C.W. and H.C.; Resources, F.W.; Software, Y.L.; Validation, L.X.; Writing—original draft, Y.L.; Writing—review & editing, Y.L. and L.X.

**Funding:** This research was funded by Fundamental Research Funds for the Central Universities of China, grant number N172608005 and Liaoning Provincial Natural Science Foundation of China, grant number 20180520007.

**Conflicts of Interest:** The authors declare no conflict of interest.

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
