# Peer review of "Topological Map Construction Based on Region Dynamic Growing and Map Representation Method"

_applsci, doi:10.3390/app9050816_

Round 1

Reviewer 1 Report

please rewrite the manuscript for sake of better understanding. the hints are as follows:

Rewriting the abstract (taking out of abbreviations: HSV..., typos: The the, and the logical structure)

introduction: what is HSV-RGB? please write in words not in abbreviation when you introduce the idea. figure 1: please use 1, 2,..., in the arrows to show the sequential order.

section 2: authors used some notation p_h, p_p, which needs to be defined beforehand or shown in Figure 2. same goes to Table 2 (algorithm 2). equation 5: we can use p(m=1) to do the job; what is the significance of the ratio.

line 239-241: please rewrite taking out the name. authors may cite a reference. equation (7): please replace t by other variable because it is already used as time.

in equation (9) we see that the authors introduced x_map and x_pixel. The authors somehow define map, trajectory, grid, grid-representing information (color, pixel), and temporal issue. Otherwise, it is difficult to follow the manuscript and evaluate the outcome.

Author Response

Dear teacher:

How are you?

Thank you very much for your review and your comments have helped me a lot.

     For details on the response, please see the attached document.

I will return the revised draft to you now. Can you help me to see other problems?

Best wishes!

Yuqiang LIU

Northeastern University

January 25, 2019

Reviewer 2 Report

Nice piece of work, you definitely need to update your reference list because right now you refer to 12 out of 28 in total (less than 50%). This is a short reflection in last 5 year's bibliography in the field. You should consider to enrich it with refs like,

‘Design of an Autonomous Robotic Vehicle for Area Mapping and Remote Monitoring’, International Journal of Computer Applications, (ISSN: 0975 – 8887), Vol. 167, No 167, June 2017

Author Response

(The authors gave the same response as above.)

Round 2

Reviewer 1 Report

the authors have improved the manuscript significantly.